# Ketoanalogues Supplemental Low Protein Diet Safely Decreases Short-Term Risk of Dialysis among CKD Stage 4 Patients

**DOI:** 10.3390/nu14194020

**Published:** 2022-09-28

**Authors:** Chieh-Li Yen, Pei-Chun Fan, Jia-Jin Chen, George Kuo, Ching-Chung Hsiao, Chao-Yu Chen, Yi-Ran Tu, Hsiang-Hao Hsu, Yung-Chang Chen, Chih-Hsiang Chang

**Affiliations:** 1Kidney Research Center, Department of Nephrology, Chang Gung Memorial Hospital, Linkou Branch, Taoyuan 33305, Taiwan; 2College of Medicine, Chang Gung University, Taoyuan 33305, Taiwan

**Keywords:** low protein diet, CKD, ESKD, ketosteril, dietary therapy

## Abstract

Background: Rigid dietary controls and pill burden make a very-low protein (0.3–0.4 g/kg body weight per day), vegetarian diet supplemented with ketoanalogues of amino acids (sVLPD) hard to follow in the long-term. This study aimed to evaluate whether a ketoanalogue supplemental low-protein diet (sLPD) (0.6 g/kg body weight per day) could also reduce the risks of dialysis among CKD stage 4 patients. Methods: Patients aged >20 years with a diagnosis of stage 4 CKD who subsequently received ketosteril treatment, which is the most commonly used ketoanalogue of essential amino acids, between 2003 and 2018 were identified from the Chang Gung Research Database (CGRD). Then, these individuals were divided into two groups according to the continuation of ketosteril for more than three months or not. The primary outcome was ESKD requiring maintenance dialysis. Results: With one-year follow-up, the continuation group (*n* = 303) exhibited a significantly lower incidence of new-onset end-stage kidney disease (ESKD) requiring maintenance dialysis (6.8% vs. 10.4%, hazard ratio [HR]: 0.62, 95% confidence interval [CI]: 0.41–0.94) in comparison to the discontinuation group (*n* = 238). Conclusions: This study demonstrated that initiating sLPDs since CKD stage 4 may additionally reduce the short-term risks of commencing dialysis without increasing CV events, infections, or mortality.

## 1. Introduction

The kidneys are known to be the main organ to clear degradation products of protein, such as p-cresyl sulfate, indoxyl sulfate, acid load, and urea. Among patients with chronic kidney disease (CKD) whose nephrons are reduced, protein load would result in the accumulation of protein degradation products and hyperfiltration of residual nephrons, and further accelerate the decline of renal function [1]. Thus, protein restriction has long been used as a treatment strategy for more than a century [2,3]. However, prolonged restriction of protein intake without nutritional supplementation may increase the risks of malnutrition [4], especially in the CKD population, which is originally vulnerable to protein energy wasting (PEW) [5,6]. The concern of malnutrition has thus limited the application of dietary therapy in the CKD population.

In 1970s, Walser et al. first demonstrated that a very low-protein diet with the supplementation of ketoanalogues of essential amino acids can safely delay the progression of CKD [7]. Theoretically, if the protein intake is well restricted, ketoanalogues of essential amino acids can utilize circulating amino groups to transfer themselves into essential amino acids through the transamination effect [8,9]. Therefore, for patients on supplemental low-protein diets (sLPDs) (0.6 g/kg body weight per day) [10,11] or supplemental very low-protein diets (sVLPDs) (0.3–0.4 g/kg body weight per day) [12,13], the supplementation of ketoanalogues of essential amino acids could reduce the risks of protein deficiency without increasing nitrogen burden. After Walser, several following randomized control trials (RCT) proved the treatments of sLPDs and sVLPDs are nutritionally safe and most participants could maintain body mass index [14,15]. Most of them also demonstrated that the dietary therapy could help preserve renal function and provide a favorable metabolic effect, including less hyperphosphatemia, less hyperlipidemia, lower proteinuria, and lower blood pressure [14,16]. However, because of short follow-up periods and small sample sizes, most RCTs were underpowered to prove the benefit of dietary therapy by using hard end points, such as new-onset end stage kidney disease (ESKD) requiring permanent dialysis and mortality. Recently, one RCT performed by Garneata et al., which enrolled 207 non-diabetic vegetarians, proved that sVLPDs could significantly reduce the requirement of renal replacement therapy in 15-months follow-up [17]. However, the strict very low- protein vegetarian diets in this study may be difficult to long-term follow by most patients, and so is it hard to promote this kind of extremely strict dietary therapy to the whole CKD population. Thus, the effect of sLPDs, which might be easier to follow long-term for most CKD patients, is still worthy of further investigation.

Indeed, one recent RCT performed by Bellizzi, V. et al., which enrolled patients with CKD stage 4 and stage 5, compared the long-term kidney and patient survival between patients that received sVLPDs and those that received LPD [18]. They found that the sVLPDs group did not have an additional advantage to the LPD group. The associated editor, T Alp Ikizler, also commented that previous studies mostly reported the benefit of sVLPDs against a normal or high-protein diet, not compared with LPD, and suggested that LPD and sVLPD might affect both renal and patient outcomes favorably [19].

Taiwan’s National Health Insurance (NHI) launched a multidisciplinary care program for pre-ESKD patients in 2006 [20], which promoted patients to receive a low-protein diet (0.6 g/kg/day) and regularly visit dietitians. For patients that could follow dietary therapy and had a serum creatinine value that exceed 6 mg/dL, ketosteril, the most commonly used ketoanalogues of essential amino acids could be prescribed without copayment. By using data from the program, several large-scale observational studies on the benefits of sLPDs in patients with CKD stage 5 have been published [21,22,23]. However, thus far, evidence of dietary therapy among CKD stage 4 patients in Taiwan is still lacking, which this current study aimed to evaluate.

## 2. Materials and Methods

### 2.1. Data Source

This research was performed by using the Chang Gung Research Database (CGRD), which is a de-identified database on the electronic medical records of the Chang Gung Memorial hospital system [24]. Chang Gung Memorial hospital system is currently the largest medical network of Taiwan, comprising 4 tertiary medical centers and 3 other teaching hospitals across different regions, and accounts for around 10% of all Taiwan’s annual medical services [24,25]. CGRD contains comprehensive medical records, including outpatient visits, medication prescriptions, inpatients orders, procedure interventions, laboratory data, and examination reports, from the Chang Gung Memorial hospital systems. Even those self-paid treatments or medications without being covered by Taiwan’s National Healthcare Insurance could also be identified from CGRD, which allowed researchers to explore the effects of newly developed treatments or off-label use of current medications. The diagnosis of diseases in CGRD is based on the International Classification of Diseases, 9th Revision, Clinical Modification (ICD-9-CM) before 2016, and ICD-10-CM thereafter. Data regarding CGRD that could identify specific patients were encrypted and scrambled to protect patients’ privacy, and the medical information was linked by using these consistent data encryptions for research purposes. Thus, this CGRD study qualified for the waiver of informed consent according to the Chang Gung Medical Foundation’s Institutional Review Board (approval number: 201900840B0).

### 2.2. Study Design

As illustrated in Figure 1, we included patients aged >20 years with a diagnosis of stage 4 CKD (estimated glomerular filtrate rate (eGFR) between 15 and 29 mL/min/1.73 m^2^) and subsequently received ketosteril treatment from CGRD between 2003 and 2018. The date of prescribing ketosteril was defined as the index date. The individuals were further divided into two groups according to the continuation of ketosteril for more than 3 months or not. Patients under maintenance dialysis or that had ever received renal transplantation before the index date were excluded from this research. Patients who initiated dialysis or passed away in the first three months after the index date were also excluded.

### 2.3. Assessment of Covariates

Covariates of this research included demographics (age, sex, and body mass index (BMI)), comorbidities, medications and laboratory data at baseline, and utilization of medical resources (number of outpatient visits on nephrology/all departments and admission or not in the previous year). The comorbidities of this study included coronary artery disease, hypertension, diabetes, atrial fibrillation, liver cirrhosis, peripheral artery disease, dementia, systemic lupus ertyhematosus, hepatitis B virus infection, and hepatitis C virus infection. The historical events included prior hospitalization for heart failure, myocardial infarction, and stroke. Baseline medications were identified according to the prescriptions received within the 90 days period preceding the index date. The baseline laboratory data, including blood urine nitrogen, serum creatinine, proteinuria, serum bicarbonate, sodium, potassium, calcium, and phosphate, lipid profile, hemoglobin, glycohemoglobin (HbA1c), uric acid, and albumin, were identified using the most recent data within 3 months preceding the index date.

### 2.4. Outcome Measures

The primary outcome of this study was ESKD requiring maintenance dialysis. The secondary outcomes were all-cause mortality, infection-related mortality, major adverse cardiac and cerebrovascular event (MACCE, defined as a composite of cardiovascular death, acute myocardial infarction (AMI), and ischemic stroke). These outcomes were identified according to the medical records of CGRD. New-onset ESKD requiring permanent dialysis was defined by receipt of a catastrophic illness certificate for exemption from medical expenditure of long-term dialysis. AMI, ischemic stroke, and infection-related death were identified based on the principal diagnosis of hospitalization or emergency room visits. The composite of study endpoint was defined as the occurrence of any primary or secondary outcomes of this study. Because those who died or initiated permanent dialysis within 3 months after index date were excluded from this study, the study outcomes were identified since the 90th day after index date. As previous studies reported, it is not easy for patients to follow up diet restriction in the long-term, so an observational study could not guarantee a long-term adherence to LPD. Thus, this study compared short-term outcomes and long-term outcomes between groups, respectively. The follow-up period was measured between the 90th day after index date and the first occurrence of any study outcome independently, until one year after the index date (1-year follow-up), or until the end of the cohort (at the end follow-up), whichever came first.

### 2.5. Statistical Analysis

The baseline characteristics of patients between the two groups (continuation vs. discontinuation) were elaborated to balance using inverse probability of treatment weighting (IPTW) based on propensity score. The propensity score was the predicted probability of being in the continuation group derived from the multivariable logistic regression analysis. All of the variables listed in Table 1 were treated as covariates in the logistic regression model, except the follow up years were replaced with the index date. To prevent the extreme weights from impacting the results, the weights over the 99th percentile were truncated [26]. In addition, the IPTW was conducted on the imputed cohort with complete data using the single expectation maximization algorithm due to the fact that there was a substantial number of missing values. The balance of the baseline characteristics between groups was assessed using the standardized difference (STD), where an absolute STD value less than 0.1 indicated negligible group difference [26].

The incidence of ESKD requiring dialysis between groups was compared using a Fine and Gray subdistribution hazard model which considered both death and renal transplantation during follow up a competing risk. The incidence of other non-fatal outcomes (e.g., acute myocardial infarction and ischemic stroke) between groups was also compared using the Fine and Gray model, where only death was considered a competing risk. The risk of fatal outcomes (e.g., cardiovascular death and composite outcome) between groups was compared using Cox proportional hazard model. Subgroup analysis was conducted on the ESKD requiring dialysis during 1-year follow up by age (20–65 vs. ≥65 years), gender, body mass index, (<24 vs. ≥24 kg/m^2^), baseline eGFR (15–20 vs. 21–30 mL/min/1.73 m^2^), hypertension, diabetes, cardiovascular disease, use of ACEi/ARB, loop diuretics and pentoxyfilline, proteinuria, albumin level (<3.5 vs. ≥3.5 mg/dL), and hemoglobin (<10 vs. ≥10 g/dL). A two-sided *p* value of <0.05 was considered significant. All analyses were conducted using SAS 9.4 (SAS Institute, Cary, NC, USA).

## 3. Results

### 3.1. Patient Characteristics

A total of 541 adult patients, who were diagnosed with CKD stage 4 and subsequently received ketosteril treatment between 2003 and 2018, were extracted from CGRD. Among them, 303 patients had been on ketosteril treatment for more than three months (the continuation group), and the other 238 patients discontinued ketosteril within three months after the first prescription (the discontinuation group). The demographics, comorbidities, medical utilization, medications, and baseline laboratory data are displayed in Table 1 and Table 2. Before applying IPTW, the continuation group exhibited younger age, greater high-density lipoprotein level, lower proteinuria, lower phosphorus, and lower serum uric acid compared to the discontinuation group (the absolute STD values >0.1). After IPTW application, all the absolute STD values were less than 0.1, which indicated that the clinical characteristics of the two groups were well balanced (Appendix A).

### 3.2. Follow-Up Outcomes

For better understanding the short-term and long-term impact of LPD, this study compared the outcomes with one-year follow-up and at the end of follow-up. As presented in Table 3, with one-year follow-up, the continuation group exhibited a significantly lower incidence of new-onset ESKD requiring maintenance dialysis (6.8% vs. 10.4%, hazard ratio [HR]: 0.62, 95% confidence interval [CI]: 0.41–0.94) compared to the discontinuation group. Regarding secondary outcomes, the risk of MACCEs (3.2% vs. 5.3%, HR: 0.59, 95% CI: 0.25–1.39), all-cause death (3.8% vs. 5.6%, HR: 0.65, 95% CI: 0.27–1.54), and infection-related death (3.1% vs. 3.6%, HR: 0.82, 95% CI: 0.29–2.33) did not significantly differ between groups. However, the continuation group still exhibited a significantly lower rate of composite of study endpoint (9.9% vs. 15.9%, HR: 0.82, 95% CI: 0.34–0.97). On the other hand, at the end of follow-up, the risk of all-cause death, MACCE, ESKD requiring dialysis, infection-related death, and composite of study endpoint did not significantly differ between the two groups. The cumulative event rates of ESKD requiring dialysis and composite of study endpoint with one-year follow-up or at the end of follow-up are presented (Figure 2 and Figure 3).

### 3.3. Subgroup Analysis

To further analyze whether the different clinical situations modified the association between the continuation of ketosteril treatment and primary outcome, we performed subgroup analyses for new-onset ESKD requiring dialysis during the one-year follow up (Table 4). The results showed that the beneficial effect of the continuation of ketosteril treatment persisted across all subgroups, with all of the non-significant interaction effects.

## 4. Discussion

As far, two RCTs have proven the impressive benefits of dietary therapy to delay or even avoid requirement of permanent dialysis [17,27]. However, these studies focused on vegetarian sVLPDs, which restricted daily protein intake within 0.3–0.4 g/kg body weight per day and patients needed to take only plant-based protein. Meanwhile, 12 tablets/day of ketosteril were required for a 60-kg adult undergoing sVLPDs. Such rigid dietary controls and pill burden make sVLPDs hard to follow long-term [28]. One recent study focusing on patients with CKD stage 4 and stage 5 also indicated that, if the adherence to diet restriction is low, prescribing sVLPDs does not provide additional advantage compared to standard LPD [18]. By contrast, Taiwan’s pre-ESKD care program promoted sLPD for CKD stage 5 patients, which restricted daily protein intake within 0.6 g/kg body weight with lower doses of ketosteril (a maximum of 6 tablets/day) and thus may be easier to follow for patients long-term. Similarly, one famous Italian dietary therapy team also used and advocated the plant-based low-protein diet supplemented with fewer ketosteril [29]. More importantly, by taking advantage of data from pre-ESKD care program, large-scale observational studies have demonstrated sLPDs could safely delay the requirement of permanent dialysis for around three months in CKD stage 5 patients and patients ever received sLPDs have better long-term outcomes after initiation of permanent dialysis, including lower CV events, infection, and all-cause mortality. However, Taiwan’s pre-ESKD care program only enrolled patients with CKD stage 5 and national healthcare insurance would not cover the payment of ketosteril until eGFR < 15 mL/min/1.73 m^2^. Patients with early stage of CKD could only receive sLPDs at their own expenses and their information of treatment could not be identified in national healthcare insurance research database. Thus, so far, there is no available evidence concerning the role of dietary therapy among CKD stage 4 patients in Taiwan.

This study was conducted on the basis of CGRD, the dataset of Chang Gung Medical System, which covered one-tenth of Taiwan’s total medical services and had comprehensive information of medical records. Most importantly, those self-paid treatments and medications could be identified from CGRD, which allowed us to enroll patients received sLPDs since CKD stage 4. However, since the costs of ketosteril and the consults of dietitians were not covered by healthcare insurance for CKD stage 4 population, the characteristics of patients who are willing to start dietary therapy since CKD stage 4 must be much different from non-users. For example, physicians may suggest self-paid dietary therapy for patients that experienced or expected a rapid decline of eGFR. If we simply compared outcomes between users and non-users, these extreme bias by indication is believed to be difficult to eliminate with statistical strategies. Instead, in this study, we enrolled all patients received sLPDs since CKD stage 4 and compared interested outcomes between patients could continue treatment for more than three months and those who discontinued it within first three months. In Table 1, the two groups exhibited only minimal differences in baseline characteristics even before propensity score weighting, which implies that this is an appropriate grouping strategy. However, by using database study, it is difficult to identify the reason why patients discontinued sLPDs. One previous Taiwanese, small-scale, phenomenological study might provide some clues [30]. By interviewing CKD patients under diet therapy, the previous study reported that the major barrier for patients to continue sLPDs concerned confusion about eating restrictions, struggling with the daily diet, concerns about quality of life, and the motivation for adherence to a low-protein diet, which implied that the discontinuance of sLPDs among CKD patients might be mainly because of intolerable dietary restrictions. On the other hand, in a study aiming to evaluate the effect of dietary therapy, it is hard to clarify if the beneficial effect is from dietary therapy itself or from the better medical adherence among patients who could tolerate the diet restriction. In this study, we tried to reduce this confounder. First, we balanced the out-patient visits of nephrology and all departments, which might be regarded as surrogate markers for medical compliance since patients with better medical compliance might need frequent out-patients visits for prescriptions. Second, we balanced commonly used medication for CKD patients, such as pentoxyfilline and ACEi/ARB. Thus, we believed, though some inherent bias must exist, the continuance of sLPDs or not was the major influence factor for different outcomes between groups in this study.

In results, even if CKD stage 4 patients who received dietary therapy are few in number, this study still observed a beneficial effect in renal outcomes in patients continuing sLPDs. Although current clinical trials of dietary therapy focus on sVLPDs with plant-based protein, several small-scale studies still exhibited that the sLPDs have benefits in some “soft endpoint” of renal outcomes, including lower proteinuria, less uremia, and better sugar control [31,32]. This study additionally demonstrated that the continuation of sLPDs in CKD stage 4 patients is associated with significantly lower short-term risks of commencing permanent dialysis. Although, in this study, the difference of renal outcomes between two groups is less apparent in long-term follow-up, we speculated the disappearance of long-term benefits in this study might be due to two major reasons. First, an observational study could not guarantee patients’ long-term compliance of dietary therapy. Second, because the costs of dietary therapy could be covered by national healthcare insurance if eGFR < 15 mL/min/1.73 m^2^, some patients discontinuing sLPDs in the first place may restart dietary therapy after CKD stage 5, which is likely to make the long-term renal outcomes between two groups less different. Anyhow, on the basis of the renal benefits of sLPDs in CKD stage 5 patients, which have been proven in several previous Taiwanese studies [11,21], our findings imply that early initiation sLPDs since CKD stage 4 could be further beneficial in delaying the requirement of dialysis. Besides, regarding safety issues, previous studies have proven that, if the daily calorie intake is sufficient, sVLPDs or sLPDs in CKD patients could be safe, and even had some metabolic benefits, namely less hyperphosphatemia, less hyperparathyroidism, lower blood pressure, and gut microbiota modulation, which may additionally contribute to long-term cardiovascular protective effect and patient survival. One previous cohort study demonstrated that serum urea level is associated with CV outcomes in CKD patients, and further suggested a reduction of urea by nutritional therapy may thus be beneficial in reducing CV risks [33]. Actually, in this study, if we combined all common adverse events in CKD patients, including ESKD requiring dialysis, MACCEs, and all-cause mortality, into a composite outcome, patients continuing sLPDs would exhibit significantly lower risks of the composite outcomes. However, due to the low number of patients enrolled, this study is insufficient to prove the benefits of sLPDs in reducing CV events, mortality, and infections, respectively. Further large-scale studies are warranted to evaluate if there are beneficial effects of sLPDs besides retarding CKD progression.

Some limitations in this study should be acknowledged. First, the major limitation is the relatively smaller number of enrollees, which resulted in underpower of this study to evaluate outcomes with relatively lower incident rate. However, the enrollees of this study are still greater in number than most of previous randomized-control trials on dietary therapy. Because diets need commitment and compliance, which is difficult to assess, a randomized control study of dietary therapy would be hardly feasible, which gave observational study an edge. Second, although CKD patients under low protein diet in the Chang Gung medical system would receive nutrition education from a dietitian, an observational study could not ensure that every enrollee would fully comply to a low-protein diet or ketoanalogues supplementation. This study is not able to provide the information of daily protein intake (self-record or calculating by urine urea) either. Third, although metabolic acidosis is an important factor for CKD progression, because routine serum bicarbonate follow-up is not required in Taiwan’s pre-ESKD care program, there are fewer available data regarding serum bicarbonate in this study. Forth, this study allocated enrollees by using the continuation of dietary therapy for more than three months or discontinuation within three months to reduce confounding by indication, but an observational study must entail some inherent limitations.

In conclusion, by using the continuation of dietary therapy or not to reduce the possible bias by indication, this study demonstrated that initiating sLPDs since CKD stage 4 may additionally reduce the short-term risks of commencing dialysis without increasing CV events, infections, or mortality. We expect that this study could engage researchers to perform more well-designed observational or interventional studies to validate our findings.

## Figures and Tables

**Figure 1 nutrients-14-04020-f001:**
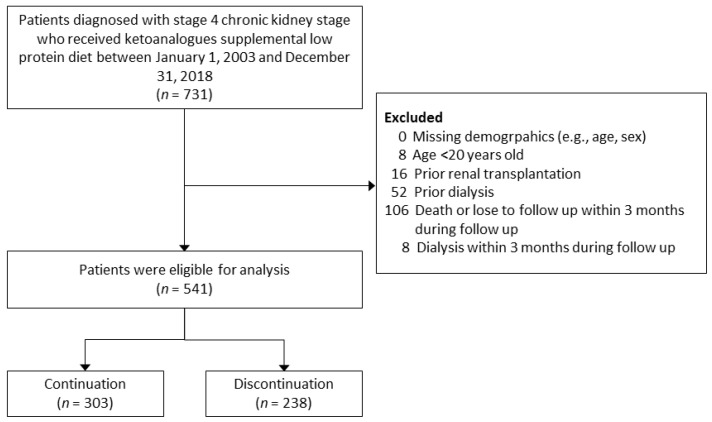
Flowchart for the inclusion and exclusion of the study patients.

**Figure 2 nutrients-14-04020-f002:**
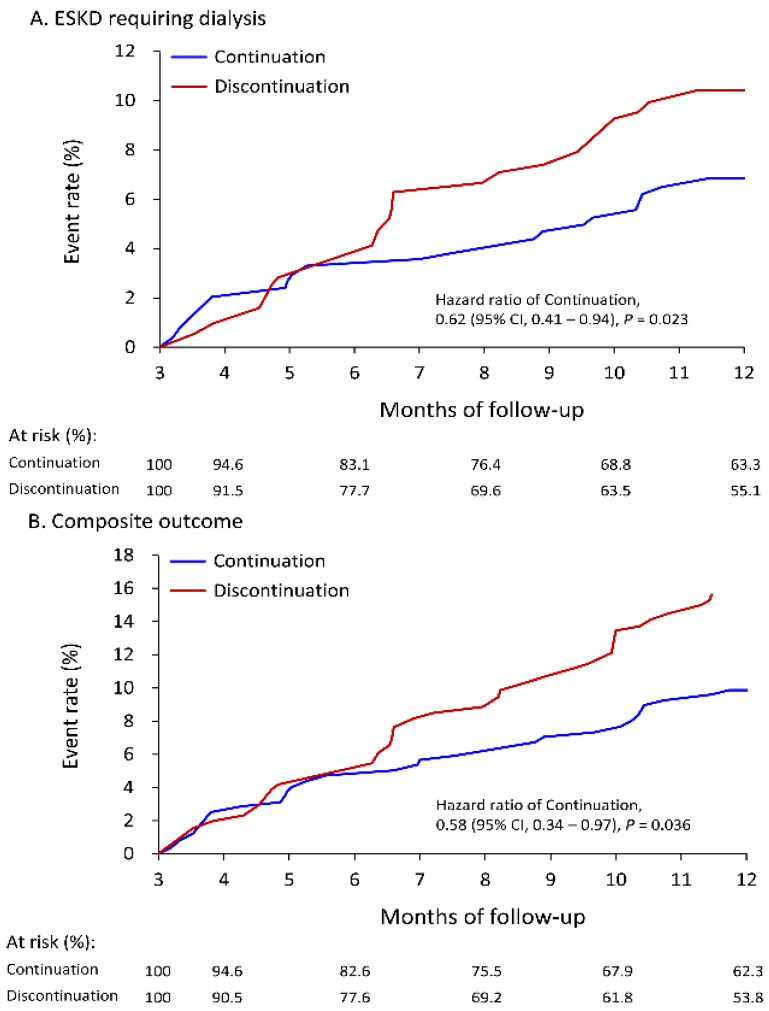
Cumulative event rate of ESKD requiring dialysis (**A**) and composite outcome (**B**) during 1-year follow up of patients receiving ketoanalogues supplemental low protein diet in the IPTW adjusted cohort. The composite outcome included anyone of ESKD requiring dialysis, cardiovascular death, acute myocardial infarction, ischemic stroke and all-cause death. ESKD, end stage kidney disease; IPTW, inverse probability of treatment weighting.

**Figure 3 nutrients-14-04020-f003:**
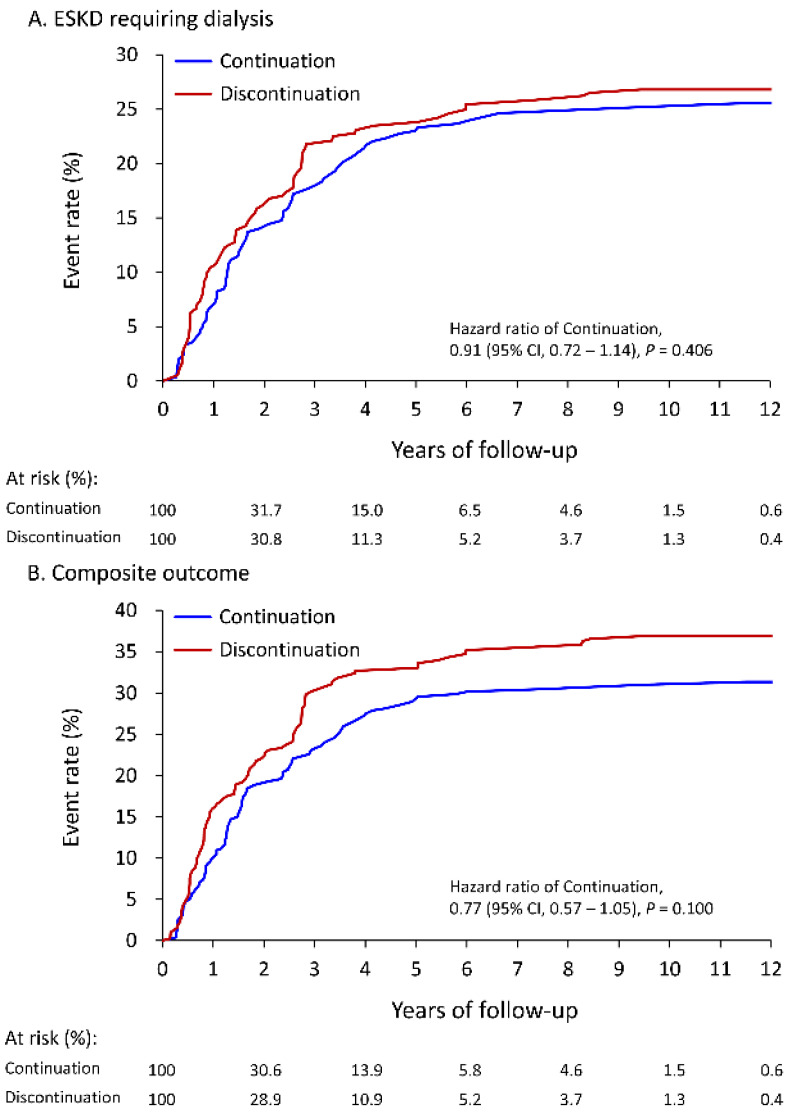
Cumulative event rate of ESKD requiring dialysis (**A**) and composite outcome (**B**) at the end of follow up of patients receiving ketoanalogues supplemental low protein diet in the IPTW adjusted cohort. The composite outcome included anyone of ESKD requiring dialysis, cardiovascular death, acute myocardial infarction, ischemic stroke and all-cause death. ESKD, end stage kidney disease; IPTW, inverse probability of treatment weighting.

**Table 1 nutrients-14-04020-t001:** Baseline characteristics of patients receiving ketoanalogues supplemental low protein diet before IPTW and EM imputation.

Variables	Available Number	Continuation (*n* = 303)	Discontinuation (*n* = 238)	STD	*p* Value
Age, years	541	67.1 ± 14.1	68.5 ± 14.4	−0.10	0.24
Age ≥ 65 years	541	171 (56.4)	157 (66.0)	−0.20	0.02
Male	541	187 (61.7)	147 (61.8)	<0.01	0.99
Body mass index, kg/m^2^	438	26.4 ± 24.2	24.8 ± 4.7	0.09	0.34
eGFR at index, mL/min/1.73 m^2^	541	20.9 ± 4.8	20.9 ± 4.6	<0.01	0.97
Comorbidities					
Coronary artery disease	541	87 (28.7)	62 (26.1)	0.06	0.49
Hypertension	541	248 (81.8)	197 (82.8)	−0.02	0.78
Diabetes mellitus	541	152 (50.2)	131 (55.0)	−0.10	0.26
Atrial fibrillation	541	18 (5.9)	16 (6.7)	−0.03	0.71
Liver cirrhosis	541	26 (8.6)	17 (7.1)	0.05	0.54
Peripheral artery disease	541	29 (9.6)	23 (9.7)	<0.01	0.97
Dementia	541	15 (5.0)	18 (7.6)	−0.11	0.21
Systemic lupus erythematosus	541	5 (1.7)	3 (1.3)	0.03	0.71
Hepatitis B infection	541	20 (6.6)	11 (4.6)	0.09	0.33
Hepatitis C infection	541	12 (4.0)	9 (3.8)	0.01	0.92
Heart failure hospitalization	541	24 (7.9)	20 (8.4)	−0.02	0.84
Myocardial infarction	541	25 (8.3)	15 (6.3)	0.08	0.39
Stroke	541	29 (9.6)	30 (12.6)	−0.10	0.26
No. of outpatient visits on nephrology in the previous year	541				0.52
0		39 (12.9)	37 (15.5)	−0.08	
1–5		174 (57.4)	139 (58.4)	−0.02	
6–10		74 (24.4)	47 (19.7)	0.11	
>10		16 (5.3)	15 (6.3)	−0.04	
No. of outpatient visits on all departments in the previous year	541	12.8 ± 9.5	13.1 ± 9.1	−0.03	0.75
Admission in the previous year	541	99 (32.7)	99 (41.6)	−0.19	0.03
Follow-up years	541	1.5 [0.8, 3.4]	1.3 [0.6, 3.2]	0.03	0.74

Abbreviation: IPTW, inverse probability of treatment weighting; EM, expectation-maximization; STD, standardized difference; eGFR, estimated glomerular filtration rate. Data were presented as frequency (percentage), mean ± standard deviation or median [25th, 75th percentile].

**Table 2 nutrients-14-04020-t002:** Medication and laboratory data at baseline of patients receiving ketoanalogues supplemental low protein diet before IPTW and EM imputation.

Variables	Available Number	Continuation (*n* = 303)	Discontinuation (*n* = 238)	STD	*p* Value
Medication at baseline					
ACEi/ARB	541	179 (59.1)	138 (58.0)	0.02	0.80
Beta-blockers	541	79 (26.1)	67 (28.2)	−0.05	0.59
Calcium-channel blocker	541	139 (45.9)	106 (44.5)	0.03	0.76
Mineralocortocoid receptor antagonis	541	23 (7.6)	21 (8.8)	−0.04	0.60
Loop diuretics	541	99 (32.7)	91 (38.2)	−0.12	0.18
Nitrates	541	40 (13.2)	31 (13.0)	0.01	0.95
Vasodilator	541	19 (6.3)	17 (7.1)	−0.03	0.69
Thiazide	541	22 (7.3)	22 (9.2)	−0.07	0.40
Antiplatelet agents	541	99 (32.7)	72 (30.3)	0.05	0.55
NSAIDs	541	31 (10.2)	31 (13.0)	−0.09	0.31
Steroid	541	40 (13.2)	39 (16.4)	−0.09	0.30
Proton pump inhibitor	541	48 (15.8)	47 (19.7)	−0.10	0.24
Insulin	541	41 (13.5)	35 (14.7)	−0.03	0.70
Oral hypoglycemic agents	541	107 (35.3)	92 (38.7)	−0.07	0.42
Pentoxyfillin	541	136 (44.9)	103 (43.3)	0.03	0.71
Sodium bicarbonate	541	37 (12.2)	20 (8.4)	0.13	0.15
Fibrate	541	18 (5.9)	13 (5.5)	0.02	0.81
Statin	541	123 (40.6)	93 (39.1)	0.03	0.72
Laboratory data at baseline					
Blood urine nitrogen, mg/dL	497	43.6 ± 17.0	42.9 ± 19.2	0.04	0.67
Creatinine, mg/dL	541	2.9 ± 0.8	2.9 ± 0.8	0.06	0.46
Proteinuria group, mg/dL	335				0.02
Negative (0–4)		38 (20.2)	14 (9.5)	0.30	
Trace (5–29)		14 (7.4)	10 (6.8)	0.03	
≥1+ (≥30)		136 (72.3)	123 (83.7)	−0.28	
CO_2_	160	22.7 ± 3.8	22.5 ± 4.2	0.03	0.83
Potassium, mg/dL	496	4.4 ± 0.7	4.4 ± 0.7	−0.05	0.55
Sodium, mg/dL	351	138.9 ± 4.2	138.6 ± 4.2	0.07	0.52
Calcium, mg/dL	429	8.9 ± 0.6	8.9 ± 0.7	0.07	0.46
Phosphorus, mg/dL	323	3.8 ± 0.7	4.0 ± 0.9	−0.19	0.10
HDL, mg/dL	174	46.2 ± 13.2	43.4 ± 12.8	0.21	0.18
LDL, mg/dL	222	79.1 ± 56.7	79.3 ± 49.4	<0.01	0.98
Total cholesterol, mg/dL	122	174.9 ± 41.2	176.3 ± 55.0	−0.03	0.87
HbA1C, %	249	7.0 ± 1.5	7.1 ± 1.7	−0.08	0.51
Albumin, mg/dL	400	3.8 ± 0.5	3.8 ± 0.6	0.05	0.62
Hemoglobin, g/dL	464	10.5 ± 1.8	10.4 ± 1.8	0.02	0.83
Serum uric acid, mg/dL	386	7.1 ± 2.0	7.5 ± 2.2	−0.21	0.04

Abbreviation: IPTW, inverse probability of treatment weighting; EM, expectation-maximization; STD, standardized difference; ACEi/ARB, angiotensin-converting enzyme inhibitors/angiotensin receptor blocker; NSAIDs, non-steroidal anti-inflammatory drugs; LDL, low-density lipoprotein; HDL, high-density lipoprotein; HbA1C, glycated hemoglobin. Data were presented as frequency (percentage), mean ± standard deviation or median [25th, 75th percentile].

**Table 3 nutrients-14-04020-t003:** Time to event outcomes of patients receiving ketoanalogues supplemental low protein diet in the IPTW-adjusted cohort.

	Continuation		Discontinuation		HR (95% CI) of	
Follow Up/Outcome	Event Rate	Incidence (95% CI) *	Event Rate	Incidence (95% CI) *	Continuation	*p* Value
**1-year follow-up**						
Primary outcome: ESKD requiring dialysis	6.8%	8.2 (6.0–10.8)	10.4%	13.1 (9.7–16.5)	**0.62 (0.41–0.94) ***	0.023
MACCE †	3.2%	3.8 (2.0–5.6)	5.3%	6.4 (4.0–8.8)	0.59 (0.25–1.39)	0.225
Cardiovascular death	2.0%	2.3 (1.0–3.7)	3.3%	4.0 (2.1–5.8)	0.57 (0.19–1.71)	0.314
Acute myocardial infarction	0.9%	1.1 (0.0–2.0)	2.5%	3.0 (1.4–4.7)	0.36 (0.09–1.48)	0.156
Ischemic stroke	0.3%	0.4 (0.0–1.0)	0.8%	0.9 (0.0–1.8)	0.44 (0.03–6.83)	0.557
All-cause death	3.8%	4.4 (3.0–6.4)	5.6%	6.7 (4.3–9.1)	0.65 (0.27–1.54)	0.329
Infection related death	3.1%	3.6 (2.0–5.3)	3.6%	4.3 (2.4–6.2)	0.82 (0.29–2.33)	0.714
Composite outcome #	9.9%	11.8 (9.0–15.0)	15.9%	20.2 (15.9–24.4)	**0.58 (0.34–0.97) ***	0.036
**At the end of follow-up**						
Primary outcome: ESKD requiring dialysis	25.6%	12.1 (10.0–14.2)	26.8%	13.9 (11.6–16.2)	0.91 (0.72–1.14)	0.406
MACCE †	11.5%	4.6 (3.0–5.8)	13.2%	5.7 (4.4–7.1)	0.81 (0.49–1.33)	0.396
Cardiovascular death	8.6%	3.3 (2.0–4.2)	9.8%	4.1 (3.0–5.2)	0.79 (0.44–1.39)	0.408
Acute myocardial infarction	3.3%	1.3 (1.0–1.9)	3.7%	1.6 (0.9–2.3)	0.83 (0.33–2.09)	0.698
Ischemic stroke	2.9%	1.2 (1.0–1.7)	3.6%	1.5 (0.8–2.2)	0.75 (0.28–2.03)	0.573
All-cause death	11.4%	4.4 (3.0–5.4)	13.9%	5.8 (4.5–7.2)	0.74 (0.45–1.21)	0.228
Infection related death	8.4%	3.2 (2.0–4.2)	9.2%	3.9 (2.8–4.9)	0.82 (0.45–1.50)	0.521
Composite outcome #	31.4%	15.3 (13.0–17.6)	36.9%	19.6 (16.9–22.3)	0.77 (0.57–1.05)	0.100

* Number of events per 100 person-years; † Anyone of cardiovascular death, acute myocardial infarction and ischemic stroke. # Anyone of the study outcomes, including ESKD requiring dialysis, MACCE and death.

**Table 4 nutrients-14-04020-t004:** Subgroup analysis of ESKD requiring dialysis during 1-year follow up in the IPTW adjusted cohort.

	Event Rate		HR (95% CI) of Continuation	*p* for Interaction §
Subgroup	Continuation	Discontinuation		
Age group				0.375
20–65	8.3%	9.0%	0.88 (0.32–2.39)	
>65	5.9%	11.3%	0.48 (0.20–1.15)	
Gender				0.186
Female	7.6%	6.5%	1.15 (0.38–3.45)	
Male	6.4%	12.7%	0.46 (0.21–1.01)	
Body mass index, kg/m^2^				0.974
<24	11.3%	18.1%	0.58 (0.27–1.27)	
≥24	4.7%	7.4%	0.60 (0.18–2.01)	
eGFR at baseline				0.843
15–20 mL/min/1.73 m^2^	10.3%	16.4%	0.60 (0.29–1.25)	
21–30 mL/min/1.73 m^2^	3.1%	4.0%	0.71 (0.18–2.71)	
Hypertension				0.934
No	4.8%	7.1%	0.67 (0.10–4.45)	
Yes	7.3%	11.1%	0.61 (0.31–1.21)	
Diabetes mellitus				0.430
No	5.7%	10.8%	0.46 (0.17–1.26)	
Yes	7.9%	10.1%	0.78 (0.34–1.78)	
Cardiovascular disease *				0.531
No	7.4%	9.4%	0.72 (0.31–1.71)	
Yes	5.9%	12.2%	0.48 (0.18–1.29)	
ACEi/ARB				0.514
No	7.2%	7.7%	0.82 (0.28–2.39)	
Yes	6.6%	12.3%	0.53 (0.24–1.17)	
Loop diuretics				0.412
No	5.6%	10.0%	0.50 (0.21–1.18)	
Yes	9.4%	11.3%	0.86 (0.32–2.31)	
Pentoxyfilline				0.719
No	7.0%	10.1%	0.69 (0.28–1.68)	
Yes	6.7%	10.8%	0.54 (0.22–1.36)	
Proteinuria				0.603
Negative/Trace	3.7%	14.2%	0.26 (0.02–3.46)	
≥1+	7.7%	13.7%	0.53 (0.24–1.16)	
Albumin, mg/dL				0.242
<3.5	15.4%	13.6%	1.02 (0.35–2.95)	
≥3.5	5.8%	12.1%	0.45 (0.18–1.10)	
Hemoglobin, g/dL				0.388
<10	10.4%	14.1%	0.66 (0.29–1.52)	
≥10	4.1%	10.7%	0.36 (0.12–1.12)	

Abbreviation: ESKD, end stage kidney disease;IPTW, inverse probability of treatment weighting; HR, hazard ratio; CI, confidence interval; eGFR, estimated glomerular filtration rate; ACEi/ARB, angiotensin-converting enzyme inhibitors/angiotensin receptor blocker; §: *p* for Interaction describes whether if the treatment effect (Continuation vs. Discontinuation) on the risk of ESKD requiring dialysis between subgroups was significantly different of not. A *p* > 0.05 of the subgroup suggests the treatment effect was not significantly different between different levels of the subgroup. *: Cardiovascular disease included history of acute myocardial infarction and ischemic stroke.

## Data Availability

The data presented in this study are available on request from the corresponding author. The data are not publicly available due to the access of Chang Gung Research Database is required the permission of Chang Gung Memorial hospital system.

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
