# Peer review of "Ketoanalogues Supplemental Low Protein Diet Safely Decreases Short-Term Risk of Dialysis among CKD Stage 4 Patients"

_nutrients, 2022, doi:10.3390/nu14194020_

Round 1
Reviewer 1 Report
Thank you for the opportunity to review this paper. In this paper, the authors have utilised linked health usage data to identify whether the continuation of ketoanalogue therapy in patients with CKD stage 4 is associated with risk of progression to requiring maintenance dialysis.
Continuation of ketoanalogue therapy was associated with a decreased risk of requiring maintenance HD at 1 year follow up compared to discontinuation, however this association was not present at the end of follow up. Overall it is a well conducted retrospective study of linked health data, with appropriate subgroup analyses performed. I have one minor point that was not adequately addressed in the discussion:
One thing that was not clearly discussed is the reasoning for discontinuation of ketosteril treatment. Given that both the continuation and discontinuation groups are relatively similar at baseline (Table 1), there does not appear to be a clinical reason that those in the discontinue group would cease treatment. Is there any insights into why the discontinue group did discontinue? I.e. due to lack of clinical need, or patient initiated discontinuation due to pill burden?
There are several typos: “researches” – line 292, 294
Author Response
Thank you for the opportunity to review this paper. In this paper, the authors have utilised linked health usage data to identify whether the continuation of ketoanalogue therapy in patients with CKD stage 4 is associated with risk of progression to requiring maintenance dialysis.
Continuation of ketoanalogue therapy was associated with a decreased risk of requiring maintenance HD at 1 year follow up compared to discontinuation, however this association was not present at the end of follow up. Overall it is a well conducted retrospective study of linked health data, with appropriate subgroup analyses performed. I have one minor point that was not adequately addressed in the discussion:
One thing that was not clearly discussed is the reasoning for discontinuation of ketosteril treatment. Given that both the continuation and discontinuation groups are relatively similar at baseline (Table 1), there does not appear to be a clinical reason that those in the discontinue group would cease treatment. Is there any insights into why the discontinue group did discontinue? I.e. due to lack of clinical need, or patient initiated discontinuation due to pill burden?
Answer: Thank the reviewer to raise this important issue. Indeed, by using a database study, it is difficult for us to identify the reason why patients discontinued sLPDs. However, one previous Taiwan’s small-scale study, by interviewing CKD patients under diet therapy, reported the major barrier for the continuation of diet therapy were confusion about eating restrictions, struggling with the daily diet, concerns about quality of life, and the motivation for adherence to low protein diet. Thus, just like the reviewer’s opinion, we speculated that these patients discontinued sLPDs were not associated with clinical reason but were due to hard to follow up diet restriction. The relevant content and reference have been added into the section of “ discussion” (line: 296)
There are several typos: “researches” – line 292, 294
Answer: Thanks for the reminder. We have modified these typos.

Reviewer 2 Report
This is a retrospective study from a database recruiting patient who received ketosteril at stage 4. Patients are separated into 2 groups depending on taking the treatment for at least 3 months. The primary outcome is time to dialysis. The authors concluded that continuation of treatment delay the time to dialysis.
Introduction and motivation of the study: the authors do not mention recent studies and reviews:
A low-protein diet supplemented with a mixture of amino acids/keto-analogues of AA may be indicated in the CKD stage 4-5 patient. In the long term, adherence to a very strict diet and adherence to the number of tablets is low. A recent randomized study (Bellizzi AJCN 2021) shows that due to these difficulties there is no advantage to sVLPD over a standard LPD. In the introduction (P2 L65) the authors should mention this study as well as the associated editorial of A.Ikizler ( 1.Bellizzi, V. et al. No additional benefit of prescribing a very low-protein diet in patients with advanced chronic kidney disease under regular nephrology care: a pragmatic, randomized, controlled trial. Am J Clin Nutrition 115, 1404–1417 (2021); 1.Ikizler, T. A. Very low-protein diets in advanced kidney disease: safe, effective, but not practical. Am J Clin Nutrition 115, 1266–1267 (2022). ) which provide arguments for a well-followed LPD. This study should be taken into account in the introduction and especially the discussion (Line 240) since 60% of the patients included in this randomized study are at stage 4.
A pragmatic approach using an LPD diet supplemented with fewer tablets (1 cp/ 10 Kg BW) has been used and advocated by Italian teams for a long time and this should be mentioned (1.Cupisti, A. et al. Medical Nutritional Therapy for Patients with Chronic Kidney Disease not on Dialysis: The Low Protein Diet as a Medication. Journal of Clinical Medicine 9, 3644–19 (2020). )
The tables are difficult to read and need to be reviewed. Table 2 should be separated (treatment and biology).
The results are not well explained, especially from Figures 2 and 3, little explained in the text: why did the authors choose to present the results at 1 year of follow-up, for patients at stage 4. This aspect need to be better specified in the results.
An important point in this context, there is no approach to diet adherence by calculating protein intake per kg of weight from urinary urea. There is mention of "blood urea nitrogen", there does not seem to be a difference between the groups, but it is not clear. Did the patients who stopped treatment returned to a richer diet?
If there is no difference, the authors should discuss the best outcome in the KA group: better overall adherence ?, specific effect of KA?. Indeed it is likely that patients who have continued beyond 3 months to take more tablets are the most observant and the most adherent to the medical instructions, which will explain the best survival without dialysis. Some results appear to be difficult to understand in the table : the authors used STD, it seems that no difference exists for most variable and the formulation could be confusing for the reader with a standard approach with t-test. Considering CO2 , only 160/541 data are available (control of acidosis is a main factor for CKD progression).
In conclusion, there is little data on the value of AA supplementation associated with a stage 4 LPD regimen. Despite the shortcomings of a retrospective study, this study provides data. Nevertheless, the discussion, in my opinion, needs to be reviewed. Tables need to be re-formatted.
Author Response
Reviewer 2
This is a retrospective study from a database recruiting patient who received ketosteril at stage 4. Patients are separated into 2 groups depending on taking the treatment for at least 3 months. The primary outcome is time to dialysis. The authors concluded that continuation of treatment delay the time to dialysis.
Introduction and motivation of the study: the authors do not mention recent studies and reviews:
A low-protein diet supplemented with a mixture of amino acids/keto-analogues of AA may be indicated in the CKD stage 4-5 patient. In the long term, adherence to a very strict diet and adherence to the number of tablets is low. A recent randomized study (Bellizzi AJCN 2021) shows that due to these difficulties there is no advantage to sVLPD over a standard LPD. In the introduction (P2 L65) the authors should mention this study as well as the associated editorial of A.Ikizler ( 1.Bellizzi, V. et al. No additional benefit of prescribing a very low-protein diet in patients with advanced chronic kidney disease under regular nephrology care: a pragmatic, randomized, controlled trial. Am J Clin Nutrition 115, 1404–1417 (2021); 1.Ikizler, T. A. Very low-protein diets in advanced kidney disease: safe, effective, but not practical. Am J Clin Nutrition 115, 1266–1267 (2022). ) which provide arguments for a well-followed LPD. This study should be taken into account in the introduction and especially the discussion (Line 240) since 60% of the patients included in this randomized study are at stage 4.
A pragmatic approach using an LPD diet supplemented with fewer tablets (1 cp/ 10 Kg BW) has been used and advocated by Italian teams for a long time and this should be mentioned (1.Cupisti, A. et al. Medical Nutritional Therapy for Patients with Chronic Kidney Disease not on Dialysis: The Low Protein Diet as a Medication. Journal of Clinical Medicine 9, 3644–19 (2020). )
Answer: Thank the reviewer for providing these important references. We have added these content in the section of “ introduction” (line: 64 ) and “ discussion”(line:262 and 268 ).
The tables are difficult to read and need to be reviewed. Table 2 should be separated (treatment and biology).
Answer: Thanks for the suggestion. For better reading, we separated the original table 1 to four parts: Baseline characteristics before IPTW (Table 1), baseline medication and laboratory data before IPTW (Table 2), Baseline characteristics after IPTW (supplemental table 1), and baseline medication and laboratory data after IPTW (supplemental table 2). The content in results have been modified accordingly.
The results are not well explained, especially from Figures 2 and 3, little explained in the text: why did the authors choose to present the results at 1 year of follow-up, for patients at stage 4. This aspect need to be better specified in the results.
Answer: Thank the reviewer to raise this important issue. As previous studies reported, it is not easy for patients to long-term follow up diet restriction, so an observational study could not guarantee a long-term adherence to LPD. Thus, this study compared short-term (1-year) and long-term (at the end of follow-up) outcomes between groups, respectively. We speculated that the short-term outcome may better reflect the benefit of good adherence to LPD. These relevant content have added in the section of “ methods” (line:147 ) and “ results” (line:210 ).
An important point in this context, there is no approach to diet adherence by calculating protein intake per kg of weight from urinary urea. There is mention of "blood urea nitrogen", there does not seem to be a difference between the groups, but it is not clear. Did the patients who stopped treatment returned to a richer diet?
Answer: Thank the reviewer to raise this important point. Because the Taiwan’s pre-ESKD care program did not request routinely test urine urea for CKD4 patients, the available data of urine urea is very limited and we thus could not estimate the protein intake per kg of weight by using urine urea. We have listed this point into the section of “limitation” (line:354 ).
If there is no difference, the authors should discuss the best outcome in the KA group: better overall adherence ?, specific effect of KA?. Indeed it is likely that patients who have continued beyond 3 months to take more tablets are the most observant and the most adherent to the medical instructions, which will explain the best survival without dialysis.
Answer: Thank the reviewer to raise this important question. This is a problem hard to solve among studies aimed to evaluate the benefit of supplemental low-protein or very-low protein diet because it is difficult to clarify the main impact is from dietary therapy or from better overall adherence. In this study, we tried our best to reduce the confounders. For example, we have listed the number of outpatient visits in nephrology or all departments, which might be a surrogate marker for medical compliance. On the other hand, we also listed commonly used medication in CKD patients. After adjusted by IPTW, these covariates are well balanced between groups.
Thus, though some inherent bias must exist, we believed that the continuance of ketoanalogue-supplemented low protein diet or not is the major influence factor for the different outcome between groups. The relevant content have added into the section of “ discussion” (line: 303)
Some results appear to be difficult to understand in the table : the authors used STD, it seems that no difference exists for most variable and the formulation could be confusing for the reader with a standard approach with t-test.
Answer: Thank the reviewer for the reminder. We have modified the table 1 and table 2 (also the supplemental table 1-2) and added the standard approach with t-test or chi-square test.
Considering CO2 , only 160/541 data are available (control of acidosis is a main factor for CKD progression).
Answer: Thanks for raising this issue. Routinely follow up CO2 is not required in the Taiwan’s pre-ESKD care program, so the available data is less. We have added this content in the section of “ limitation” (line:354 )
In conclusion, there is little data on the value of AA supplementation associated with a stage 4 LPD regimen. Despite the shortcomings of a retrospective study, this study provides data. Nevertheless, the discussion, in my opinion, needs to be reviewed. Tables need to be re-formatted.
Answer: Thank the reviewer for making very important suggestions and providing important references. We have modified our discussion and re-formatted the table accordingly.
